# MeaeQ: Mount Model Extraction Attacks with Efficient Queries

**Chengwei Dai**[1,2]**, Minxuan Lv**[1,2]**, Kun Li**[1*]**, Wei Zhou**[1]**,**
[1]Institute of Information Engineering, Chinese Academy of Sciences
[2]School of Cyber Security, University of Chinese Academy of Sciences
{daichengwei, lvminxuan, likun2, zhouwei}@iie.ac.cn

## Abstract

We study model extraction attacks in natural language processing (NLP) where attackers aim to steal victim models by repeatedly querying the open Application Programming Interfaces (APIs). Recent works focus on limited-query budget settings and adopt random sampling or active learning-based sampling strategies on publicly available, unannotated data sources. However, these methods often result in selected queries that lack task relevance and data diversity, leading to limited success in achieving satisfactory results with low query costs. In this paper, we propose MeaeQ (**M**odel **e**xtraction **a**ttack with **e**fficient **Q**ueries), a straightforward yet effective method to address these issues. Specifically, we initially utilize a zero-shot sequence inference classifier, combined with API service information, to filter task-relevant data from a public text corpus instead of a problem domain-specific dataset. Furthermore, we employ a clustering-based data reduction technique to obtain representative data as queries for the attack. Extensive experiments conducted on four benchmark datasets demonstrate that MeaeQ achieves higher functional similarity to the victim model than baselines while requiring fewer queries. Our code is available at https://github.com/C-W-D/MeaeQ.

## 1 Introduction

The adoption of Machine Learning as a Service (MLaaS) via APIs has introduced a new security challenge known as model extraction attacks (Tramèr et al., 2016). Attackers repeatedly access the API to acquire outputs by utilizing meticulously crafted inputs (queries) and subsequently train a local model based on the gathered input-output pairs. This attack aims to obtain an extracted model that closely approximates the performance of the victim model, thereby posing a substantial threat to the intellectual property of the model owner.

In black-box scenarios, wherein the architecture and training data of the victim model remain unknown, the primary concern for attackers is to consider **how to design high-quality queries with limited query budgets**. This is crucial since frequent API calls not only result in considerable costs (Correia-Silva et al., 2018) but also carries the potential of triggering the victim model's defense mechanisms (Papernot et al., 2017; Juuti et al., 2019; Zhang et al., 2021), thereby diminishing the effectiveness of the attack.

Some studies (Orekondy et al., 2019; Xu et al., 2022; Karmakar and Basu, 2023) sample queries from annotated data associated with the victim model training data, e.g., using BBC News data to extract models trained on AGNews (Zhang et al., 2015), which deviates from the assumption of black-box model extraction, where the training data distribution of the victim model is unknown. Therefore, recent research has focused on leveraging publicly available unannotated data sources for model extraction attacks. The pioneering work by Pal et al. (2020) explores the utilization of these data sources and proposes an active learning-based sampling strategy that dynamically adjusts query selection based on self-feedback. Likewise, Krishna et al. (2020) presents two query construction methods, one involving sentences combined with random words and the other involving actual sentences randomly sampled from Wikipedia. Despite the demonstrated efficacy of these methods in their individual studies, we experimentally find that the queries sampled by these approaches often suffer from category imbalance. For instance, when applying the random strategy to an online hate speech detection API, the ratio of positive to negative samples becomes skewed as high as $30 : 1$. Such a notable imbalance in sample distribution presents challenges for model training, especially when considering low query costs. We attribute this issue to task-irrelevant text content and insufficient data

---

[*]Kun Li is the corresponding author.

diversity within the queries.

In this paper, we propose MeaeQ (**M**odel **e**xtraction **a**ttack with **e**fficient **Q**ueries), a straightforward yet effective method to address these issues. MeaeQ comprises two modules: Task Relevance Filter (TRF) and Data Reduction based on Clustering (DRC). The TRF module aims to select data that is highly relevant to the task. To achieve this, we utilize a pre-trained zero-shot sequence inference classifier, in combination with API service information, to infer the entailment relations between the pre-designed prompt and all actual sentences extracted from a publicly available text corpus. The second module, DRC, is designed to mitigate information redundancy within the query pool filtered by the TRF module. To accomplish this, DRC initially extracts embeddings from all texts in the query pool and then employs a clustering method to create multiple clusters. It subsequently selects the data nearest to the centroid of each cluster as the ultimate query. Finally, we send these queries to the victim model and then use the outputs as labels to fine-tune our local model.

Extensive experiments on simulated victim models demonstrate that the model extracted by MeaeQ exhibits higher functional similarity to the victim model than the baseline methods on four benchmark datasets. Furthermore, we validate the generalizability of MeaeQ across diverse model architectures. In-depth analyses reveal the significant contributions of both the TRF and DRC in the model extraction attack. The primary contributions of this paper can be summarized as follows:

- We employ a zero-shot sequence inference classifier, combined with API service information to filter data with high task relevance.

- We design a data reduction technique based on the clustering method to alleviate the information redundancy problem in text sampling.

- Extensive experiments confirm the effectiveness of our method in model extraction attacks at a low query cost. Additionally, the queries sampled by our approach enhance the stability of the extracted model during training.

## 2   Related Work

We introduce related work from four perspectives of model extraction attacks, including the type of victim model, the type of API feedback, the query source, and the strategy of query sampling.

**Type of Victim Model.** Some works focus on machine learning models (Tramèr et al., 2016; Chandrasekaran et al., 2020) while others concentrate on neural networks (Shi et al., 2017; Milli et al., 2019), but most of which pay attention to the computer vision (CV) (Orekondy et al., 2019; Wang et al., 2020; Zhou et al., 2020; Gong et al., 2021; Wang et al., 2022). In contrast, our study targets pre-trained language models (PLMs) such as BERT same as Krishna et al. (2020).

**Type of API Feedback.** Several studies consider using the complete probability vectors of the victim model on all or top-k classes returned by the API as feedback for the query, which is less practical in a public API (Tramèr et al., 2016; Orekondy et al., 2019). Instead, following (Pal et al., 2020; Wang et al., 2022), we focus on the most challenging scenario where the API only provides the predicted hard label.

**Query Source.**   Most studies utilize query sources derived from public problem domain-related datasets (Papernot et al., 2017; Xu et al., 2022; Karmakar and Basu, 2023). In contrast, following Pal et al. (2020), we use a large public text corpus as the query source, ensuring no overlap with the private dataset of the victim model.

**Query Sampling Strategy.** In CV, query sampling strategies can be broadly categorized into three groups. Some studies employ reinforcement learning techniques (Orekondy et al., 2019), some utilize adversarial example generation methods (Papernot et al., 2017; Juuti et al., 2019; Yu et al., 2020), and some employ model inversion techniques (Gong et al., 2021). However, applying these methods to NLP is challenging due to the discrete nature of text data, in contrast to the continuous nature of image data. In NLP, the earliest work comes from Pal et al. (2020), who adopt the active learning method to iteratively sample queries. Krishna et al. (2020) construct a nonsensical sequence of words or randomly sample actual sentences as queries. Different from them, we improve the sampling strategy by task driving and information redundancy minimization, thus better playing the value of the limited query.

## 3   Methodology

In this section, we present our method for NLP model extraction attacks. We first formalize the problem and then elucidate how to sample queries with high task relevance and low information redundancy.

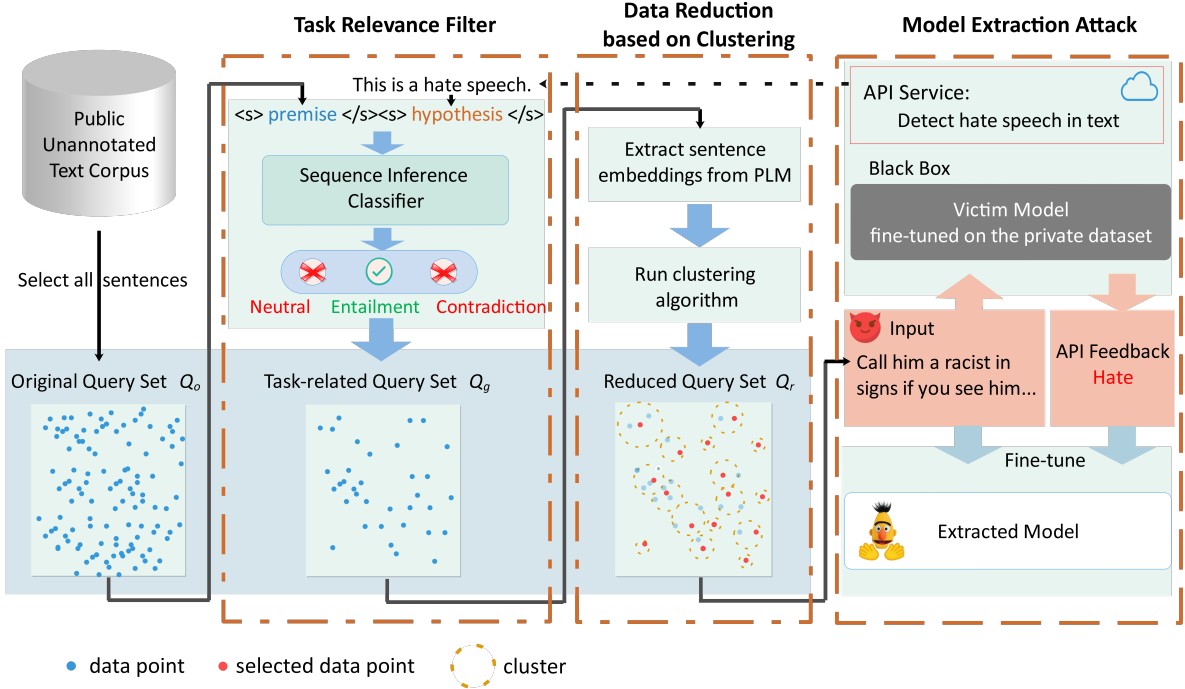

Figure 1: Overview of MeaeQ for NLP model extraction attacks. The attackers first build an original query set $\mathbf{Q_o}$ from a large text corpus. Then the attackers apply the Task Relevance Filter on $\mathbf{Q_o}$ to get a task-related query set $\mathbf{Q_g}$. Subsequently, the attacker exploits the Data Reduction based on Clustering to reduce $\mathbf{Q_g}$ to $\mathbf{Q_r}$. Finally, the attacker samples the queries from $\mathbf{Q_r}$, sends them to the API, and then uses the outputs as labels to fine-tune their own model such as BERT (Devlin et al., 2019).

## 3.1 Problem Formulation

Let $f_v$ denote the victim model ($\theta_v$ are all parameters included), which represents a black-box API providing services for the task $T$. We assume the API only returns the predicted hard label rather than probability scores. $f_v$ is trained on a private dataset that is inaccessible to the public. The attackers construct a query set and utilize the API to obtain outputs by sending sampled queries. This process generates the attacker's dataset $\{x_i, f_v(x_i)\}_{i=1}^k$, where $k$ denotes the numbers of query. Then attackers use the dataset to train their own model $f_a$ ($\theta_a$ represents all parameters included).

For evaluation, we adopt the same metrics as Krishna et al. (2020), namely *Accuracy* and *Agreement*. *Accuracy* measures the prediction accuracy of $f_a$, while *Agreement* assesses the functional similarity between $f_a$ and $f_v$. Both metrics are calculated on the private test dataset $\mathbf{D^{test}} = \{(x, y) \mid x \in \mathbf{X^{test}}, y \in \mathbf{Y^{test}}\}$ of the victim model. *Agreement* is defined as:

$$Agreement(f_v, f_a) = \frac{1}{|\mathbf{X^{test}}|} \sum_{x \in \mathbf{X^{test}}} \mathbf{I}[f_v(x) = f_a(x)] \tag{1}$$

where $\mathbf{I}(\cdot)$ is the indicator function. The calculation of *Accuracy* is equivalent to that of *Agreement*, with the only difference being the substitution of $f_v$ with the ground truth label.

Our goal is to construct a high-quality query set with a given budget to train a thief model with performance close to the victim model. The selection of queries directly impacts the parameters of $f_a$, so we can formally express the objective as:

$$\hat{\theta}_a = \arg\max_{\theta_a} Agreement(f_v, f_a) \tag{2}$$

where $\theta_v$ is fixed and hidden.

## 3.2 Overview of MeaeQ

In this subsection, we provide an overview of MeaeQ. The attackers start by sampling all actual sentences from a text corpus to initialize the original query set $\mathbf{Q_o} = \{x_i\}_{i=1}^p$, where $p$ is the number of actual sentences in the corpus. Then they utilize a sequence inference classifier to filter the input pairs that contain a hypothesis (manually designed prompt) and a premise selected from $\mathbf{Q_o}$, resulting in a task-related query set $\mathbf{Q_g}$. Next, the attackers employ a clustering-based data reduction technique

on $\mathbf{Q_g}$ to obtain a query set with low information redundancy, denoted as $\mathbf{Q_r}$. Finally, the attackers perform a model extraction attack using $\mathbf{Q_r}$. The overview of MeaeQ is illustrated in Figure 1.

### 3.3 Task Relevance Filter

Inspired by the study of Yin et al. (2019) who propose using a PLM trained on natural language inference task as a zero-shot sequence classifier for text classification, we introduce the Task Relevance Filter (TRF) combined with API service information to filter queries related to the target task. The TRF consists of a sequence inference classifier, which is a pre-trained language model trained on the MNLI dataset (Williams et al., 2018). We denote this classifier as $g$ whose input pairs are a premise and a hypothesis. We design prompt $h$ as one of the inputs (hypothesis) based on the service information of the task $T$ [1] and take the actual sentence $x_i$ from $\mathbf{Q_o}$ as another input (premise) to the model $g$ for reasoning about the relationship between them. For example, if the task is to detect if a text contains hate speech, we design the prompt as "This is a hate speech" to filter queries that resemble hate speech. Finally, we obtain the probability vector of a logical relation classification result of the actual sentence $x_i$ with prompt $h$ by the output of the model:

$$\{p_i^y\} = g\left(x_i, h\right) \tag{3}$$

where $p_i^y$ is the probability at relationship label $y \in \{neutral, entailment, contradiction\}$.

To ensure the selection of high-quality task-related queries, the Task Relevance Filter incorporates a filtering mechanism. We simply design this mechanism as follows: if $p_i^{entailment}$ is greater than or equal to a threshold $\epsilon$, the sample $x_i$ is retained, otherwise discarded. This operation is repeated until all samples in $\mathbf{Q_o}$ are classified, resulting in a task-related query set $\mathbf{Q_g} = \left\{ x_i \mid \forall x_i \in \mathbf{Q_o}, p_i^{entailment} \geq \epsilon \right\}$.

### 3.4 Data Reduction based on Clustering

To reduce the information redundancy in the query set $\mathbf{Q_g}$, we propose a Data Reduction technique based on Clustering (DRC). We transform the information redundancy problem into a graph-theoretic problem. Given a weighted undirected graph, $G = (V, E)$ where $V$ is the set of vertices and

[1] Empirically, we find that prompt template "This sentence is about [TASK_TOPIC]." works well.

$E$ is the set of edges, we extract the sentence embedding of the samples $x_i \in \mathbf{Q_g}$ as the vertex representation by the model $g$:

$$v_i = g_{embedding}\left(x_i, \varnothing\right) \tag{4}$$

Here, $v_i$ is a vector of dimension $d$. We define the edge weights $e_{i,j}$ as the cosine similarity distance between the vertices $v_i$ and $v_j$:

$$\begin{aligned} e_{i,j} &= 1 - sim_{cos}\left(v_i, v_j\right) \\ &= 1 - \frac{v_i \cdot v_j}{\|v_i\| \cdot \|v_j\|} \end{aligned} \tag{5}$$

This results in $V = \{v_i\}_{i=1}^{|\mathbf{Q_g}|}$ and $E = \left\{e_{i,j}\right\}_{i,j=1}^{|\mathbf{Q_g}|}$.

The objective of this module is to select the most representative samples from $\mathbf{Q_g}$ to form a reduced query set $\mathbf{Q_r}$. The selection problem can be viewed as finding a subgraph $\hat{G}_r$ that maximizes the sum of edge weights:

$$\hat{G}_r = \arg\max_{G_r \subset G} \sum_{e_{i,j} \in E_r} e_{i,j} \tag{6}$$

However, this problem is an NP-hard problem (already proved by He et al. (2021)). Therefore, we design an approximate solution that achieves the sample selection within a reasonable time. The steps are as follows:

**Step 1** First, we apply a clustering algorithm on the query set $\mathbf{Q_g}$ for $t$ iterations. The number of clusters $k$ is set as $|\mathbf{Q_r}|$, which represents the number of queries used to access the victim model's API. After clustering, we obtain a set of $k$ clusters $\mathbf{CLSTR} = \{clstr_i\}_{i=1}^k$ and a set of $k$ clusters centroids $\mathbf{CTRID} = \{ctrid_i\}_{i=1}^k$, where $clstr_i$ represents the set of samples in the $i$-th cluster, and $ctrid_i \in \mathbb{R}^d$ is the centroid of the $i$-th cluster.

**Step 2** For the $i$-th cluster $clstr_i$, add a sample point $x_i^r$ to the reduced query set $\mathbf{Q_r}$, satisfying the condition that $x_i^r$ is the closest to the centroid $ctrid_i$ within the cluster, i.e., the cosine similarity distance between them is the smallest:

$$x_i^r = \arg\min_{x_i \in clstr_i} 1 - sim_{cos}\left(x_i, ctrid_i\right) \tag{7}$$

**Step 3** Repeat step 2 $k$ times until all clusters are traversed, completing the construction of $\mathbf{Q_r}$.

The idea behind this approach is that the clustering algorithm generates $k$ clusters with large inter-cluster distances and small intra-cluster distances. We then select the sample point closest to

its centroid within each cluster as a candidate sample. This strategy aims to achieve an approximate maximum distance between all candidate samples.

**Time complexity analysis:** Step 1 requires $O\left(|\mathbf{Q_g}|kt\right)$ to perform clustering with $t$ iterations in total. Each iteration involves searching the smallest distance within the cluster, resulting in a total time complexity equivalent to traversing the entire set $\mathbf{Q_g}$ $t$ times, with a time complexity of $O\left(|\mathbf{Q_g}|t\right)$. Therefore, the overall time complexity of DRC remains $O\left(|\mathbf{Q_g}|kt\right)$.

# 4 Experiments

In this section, we conduct extensive experiments to evaluate the effectiveness of our method.

## 4.1 Experiment Settings

**Datasets of Victim Model**  We train simulated victim models respectively for four tasks including a hate speech detection dataset: Hate Speech (de Gibert et al., 2018), a topic classification dataset: AG News (Zhang et al., 2015) and two sentiment classification datasets SST-2 (Socher et al., 2013) and IMDB (Maas et al., 2011). Details about the data division and statistical information of these datasets can be found in Appendix B.

**Corpus and Prompts**  For query source, we use WikiText-103 corpus (Merity et al., 2017), ensuring that there is no overlap with the private datasets. We design prompts in TRF for each dataset. Since SST-2 and IMDB are both sentiment classification tasks for movie reviews, we use the same prompt "This is a movie review." for both. For AG News, which focuses on news topics, we use the prompt "This is a news.". In the case of Hate Speech, we use the prompt "This is a hate speech".

**Implementation Details**  In our experiment, we use BERT$_{\text{Base}}$ as the architecture for both the victim model and the extracted model. Additionally, we explore the effectiveness of MeaeQ on different model architectures in section 4.4. The victim model is trained for 3 epochs and the extracted model is trained for 10 epochs. We select the best checkpoints based on the validation set. We utilize BART$_{\text{Large}}$(Lewis et al., 2020) as the sequence inference classifier[2]. We use two evaluation metrics *Agreement* and *Accuracy*, as described in subsection 3.1. For each dataset, we set up several groups with different query budgets expressed as

query rates, which represent the proportion of the original dataset size. The threshold $\epsilon$ in TRF and the number of iterations $t$ in DRC are set as 0.95 and 300 respectively. We employ the faiss library[3] (Johnson et al., 2019) to accelerate the vector retrieval and use the k-means algorithm (MacQueen, 1967) for clustering. All experiments are repeated 10 times with different random seeds on a single 32GB NVIDIA V100 GPU. More details about the hyperparameters can be found in Appendix C.

## 4.2 Baselines

We compare MeaeQ with the following baselines:
**RS (Random Sampling)**  The RS is proposed by Krishna et al. (2020) [4], which randomly samples real sentences from the WikiText-103 corpus.
**AL-RS**  The AL-RS is a simple variant of active learning, which is introduced by Pal et al. (2020). In each iteration, AL-RS utilizes a random strategy. The key difference between AL-RS and RS lies in the multiple rounds of sampling versus a single round of sampling.
**AL-US**  The AL-US is proposed by Pal et al. (2020), which is also an active learning-based approach. In each iteration, the AL-US uses an uncertainty strategy that selects the top-k samples with the highest entropy value computed from the predicted probability vector of the attackers' model.

## 4.3 Main Results

In this subsection, we compare MeaeQ with the baselines on the simulated victim models trained on four datasets. The results are presented in Table 1 and Table 6 (due to space limitations, all accuracy results are listed in Appendix A). Table 1, 6 show that MeaeQ consistently outperforms the baselines for all datasets with low query budgets, demonstrating its effectiveness. Especially in Group A, we find that MeaeQ significantly outperforms the top-performing baseline on three datasets, but with slightly smaller gains on AG News. We attribute this to the similarity in data distribution between Wikipedia and AG News, as many Wikipedia articles can be considered as a form of news. We also notice that the baselines exhibit a high standard deviation in performance, implying some noise in the sampled queries. In contrast, MeaeQ can sample task-related and informative queries, leading to

---

[2]https://huggingface.co/facebook/bart-large-mnli

[3]https://github.com/facebookresearch/faiss

[4]They also propose a query construction method in which random words form sentences, but we experimentally find that this method does not work well.

| Query Budget | Method | Hate Speech ( × 1 = 1914 ) | SST-2 ( × 1 = 67349 ) | IMDB ( × 1 = 40000 ) | AG News ( × 1 = 120000 ) |
|---|---|---|---|---|---|
| Group A × 0.1 / × 0.003 (191 / 201 / 120 / 360) | RS | 45.2 ± 0.0 (45.2) | 67.8 ± 7.7 (80.7) | 53.4 ± 3.8 (63.1) | 75.9 ± 3.4 (79.6) |
| | AL-RS | 44.3 ± 2.4 (45.2) | 73.9 ± 6.9 (85.9) | 55.2 ± 4.5 (64.8) | 81.1 ± 2.7 (84.5) |
| | AL-US | 53.2 ± 7.7 (68.6) | 74.1 ± 8.1 (83.6) | 56.3 ± 4.4 (62.9) | 76.3 ± 7.4 (84.6) |
| | MeaeQ | **75.8 ± 4.5** (**79.7**) | **82.5 ± 3.6** (**86.7**) | **65.3 ± 9.1** (**79.8**) | **81.9 ± 3.3** (**86.0**) |
| Group B × 0.2 / × 0.005 (382 / 335 / 200 / 600) | RS | 45.1 ± 0.1 (45.2) | 76.6 ± 6.4 (85.4) | 59.8 ± 7.8 (71.8) | 74.0 ± 10.4 (86.2) |
| | AL-RS | 46.0 ± 3.5 (56.3) | 81.6 ± 4.2 (86.1) | 59.0 ± 7.5 (74.6) | 82.4 ± 3.8 (86.3) |
| | AL-US | 67.1 ± 7.0 (77.6) | 78.5 ± 7.6 (85.9) | 62.1 ± 8.2 (76.5) | 83.0 ± 3.7 (87.6) |
| | MeaeQ | **79.1 ± 3.0** (**81.4**) | **84.5 ± 2.4** (**87.0**) | **73.4 ± 4.9** (**78.6**) | **84.5 ± 1.7** (**87.8**) |
| Group C × 0.3 / × 0.008 (574 / 536 / 320 / 960) | RS | 47.7 ± 5.2 (61.7) | 78.6 ± 8.0 (87.5) | 63.2 ± 8.0 (73.5) | 82.3 ± 3.0 (86.7) |
| | AL-RS | 51.0 ± 9.2 (68.0) | 84.0 ± 3.9 (88.4) | 59.8 ± 6.0 (69.2) | 80.0 ± 2.2 (82.2) |
| | AL-US | 78.7 ± 6.4 (**84.3**) | 82.8 ± 5.2 (**89.9**) | 72.6 ± 5.5 (78.5) | 84.0 ± 0.9 (85.7) |
| | MeaeQ | **81.0 ± 2.6** (83.0) | **86.5 ± 1.5** (88.9) | **79.9 ± 2.0** (**82.5**) | **85.3 ± 1.8** (87.5) |

Table 1: Model extraction result (Agreement, %) under different query budgets. × 0.1 / × 0.2 / × 0.3 query budgets are set for Hate Speech dataset. × 0.003 / × 0.005 / × 0.008 query budgets are set for others. The specific number of queries for each dataset is shown in parentheses below the query budgets. The results presented in the table are the mean and standard deviation while the max value is shown in green.

higher functional similarity and more stable performance, even with extremely low query budgets.

To further validate the effectiveness of MeaeQ, we conduct experiments on more query budgets and present the results in Figure 3. We find that MeaeQ outperforms baselines in nearly all settings and exhibits lower standard deviations. In particular, for SST-2 and IMDB, MeaeQ's performance at query budget × 0.008 is comparable to or significantly surpasses the highest performance achieved by the baseline method at query budget × 0.02. When achieving the similar performance, MeaeQ reduces the query cost by more than half.

## 4.4 Cross Model Extraction

We explore the applicability of MeaeQ to different extracted / victim model architectures, including BERT$_{Base}$, RoBERTa$_{Base}$ (Liu et al., 2019), and XLNet$_{Base}$ (Yang et al., 2019), as depicted in Figure 2 and Figure 7. It is evident that MeaeQ surpasses the baselines across different model architectures, demonstrating its effectiveness and robustness. Besides, we notice that the baselines exhibit good performance solely on the matching model architecture, as indicated by the darker color on the diagonal of the heatmap. In contrast, MeaeQ consistently exhibits superior performance across different architectures, indicating its model-agnostic nature. In summary, MeaeQ can be successfully applied to various model architectures for extraction, while maintaining exceptional performance.

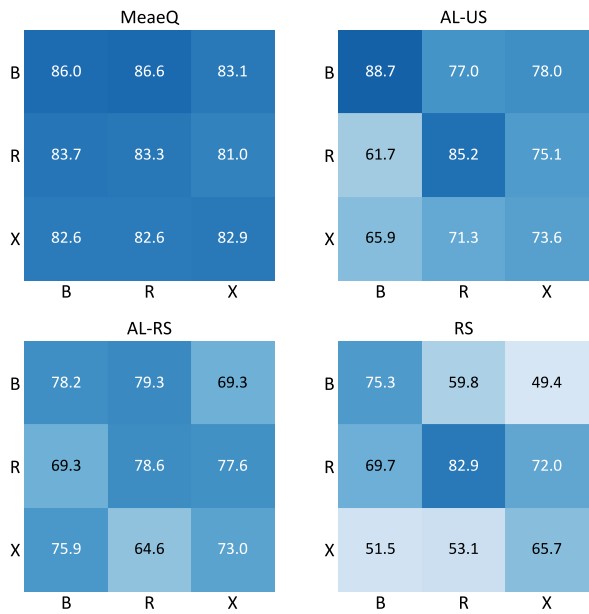

Figure 2: Cross model extraction results (Agreement, %) on Hate Speech at query budget × 0.5. The horizontal / vertical axes represent victim / extracted model architecture respectively. 'B', 'R', and 'X' represent BERT$_{Base}$, RoBERTa$_{Base}$, and XLNet$_{Base}$. Darker colors represent larger values of Agreement.

## 4.5 Ablation Study

To better understand the two key modules in MeaeQ, we compare MeaeQ with two variants, w/o TRF and w/o DRC, respectively where the corresponding modules are removed from MeaeQ. The results are shown in Figure 4 and Figure 8.

From the figures, MeaeQ apparently outperforms

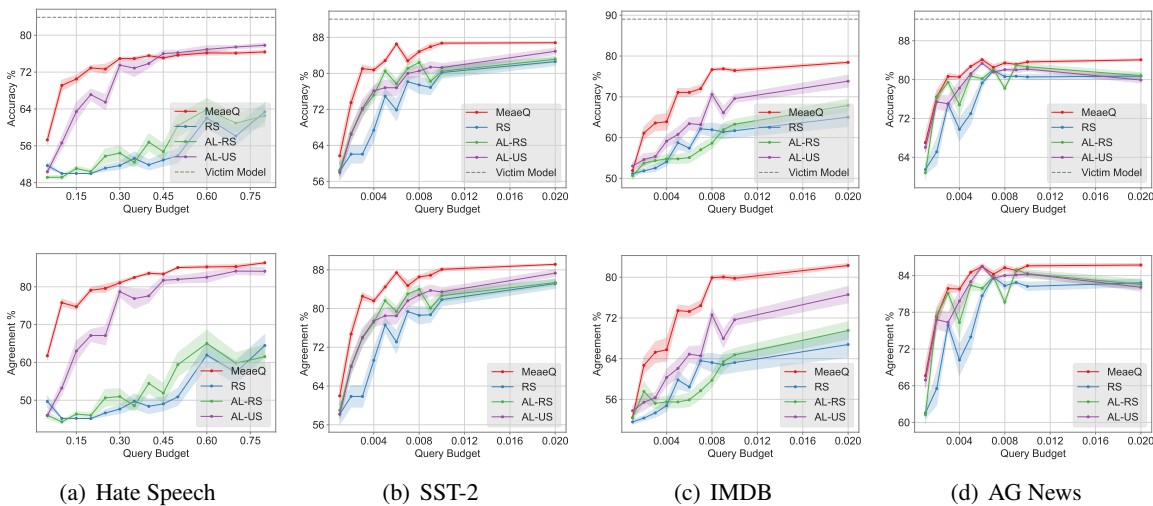

(a) Hate Speech     (b) SST-2     (c) IMDB     (d) AG News

Figure 3: Comparison between MeaeQ and the baselines at more query budgets. The first row represents the results of the Accuracy (%) on the four datasets while the second row represents the results of the Agreement (%).

the two variants, particularly in terms of higher agreement and shorter error bars at low query budgets (e.g., $\times$ 0.003 / 0.005). These results highlight the importance of TRF and DRC, as they enhance the efficacy of extraction and its stability. Moreover, MeaeQ consistently outperforms w/o TRF across all query budgets, emphasizing the critical role of task-related queries screened by TRF which aligns the attackers' data distribution with the victim model training data distribution.

However, we also observe that as the query budget increases, the performance gap between w/o DRC and MeaeQ narrows, indicating a decreasing effect of w/o DRC. We know that the query budget determines the number of clusters in DRC. As the query budget increases, the number of clusters gradually rises when the size of the query set screened by TRF remains constant. In the extreme case where the query budget equals the size of the candidate query set, the clustering algorithm becomes completely ineffective. Therefore, our method is particularly suitable for scenarios with a low query budget. However, for higher query budgets, we recommend evaluating the specific circumstances before deciding whether to utilize DRC, as DRC may degrade into random sampling under the worst-case scenario.

## 4.6 Attack on ChatGPT-based Victim Model

In this subsection, we conduct model extraction attacks on the ChatGPT-based[5] (OpenAI, 2023)

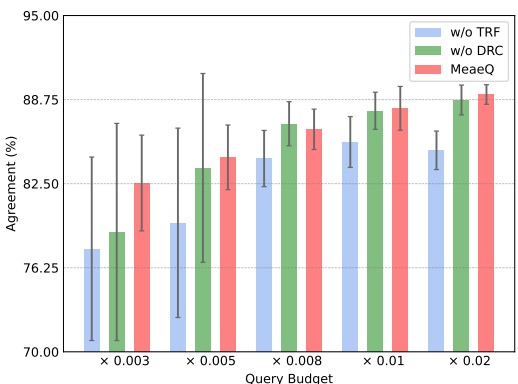

Figure 4: Ablation study results (Agreement, %) on SST-2 dataset. The horizontal axes represent different query budgets. Error bars represent standard deviation.

victim model simulated on the hate speech detection task using manual instructions to verify the effectiveness of our method in the autoregressive language model extraction. The instruction template is available in Table 2. We set up several types of extracted models with different parameters and architectures, including GPT-2 Radford et al. (2019), RoBERTa Liu et al. (2019). Comprehensive results can be found in Table 3. An analysis of the table reveals that our approach performs admirably, even when the victim model is ChatGPT-based and the extracted model relies on an autoregressive framework (e.g., GPT-2). For instance, GPT-2$_{Small}$ achieves **79.5%** functional similarity (Agreement) and 75.3% test accuracy

[5]Specifically, we call the *gpt-3.5-turbo* Chat completions API for responses. The document page is https://platform.

openai.com/docs/models/gpt-3-5

| Task | Instruction |
|------|-------------|
| Hate Speech | You are now working as a hate speech detector. Next, I will give you [batch_size] sentences. You need to indicate whether it contains hate speech. If it contains hate speech, output "Hate", otherwise output "Nohate". The required format is one output per line and the beginning of each output is numbered sequentially. Sentences:
1. [sentence_1]
2. [sentence_2]
... |

Table 2: Instruction Template. The [batch_size] in the instruction represents the number of queries, and [sentence_1], [sentence_2], etc. represent the specific query content.

(**90.4%** of the victim model). We also observe that the RoBERTa series of models outperforms the GPT-2 series in this detection task, which is clear, as masked language models are better suited for extracting models in classification tasks, while autoregressive models are typically employed for extracting generative models.

| method | GPT-2$_{\text{Small}}$ 117M | GPT-2$_{\text{Medium}}$ 345M | RoBERTa$_{\text{Base}}$ 125M | RoBERTa$_{\text{Large}}$ 355M |
|--------|------|------|------|------|
| RS | 50.0 / 43.7 | 50.2 / 43.9 | 55.6 / 51.9 | 64.4 / 59.8 |
| AL-RS | 50.2 / 43.9 | 53.6 / 48.9 | 50.0 / 43.7 | 56.3 / 54.2 |
| AL-US | 50.2 / 43.9 | 50.6 / 45.6 | 60.7 / 57.3 | 73.6 / 76.6 |
| MeaeQ | **75.3 / 79.5** | **71.5 / 75.3** | **77.4 / 82.4** | **79.5 / 85.4** |

Table 3: Model extraction results (Accuracy / Agreement, %) on ChatGPT simulated on the Hate Speech dataset at query budget × 0.8 (about 1531 queries). The accuracy of ChatGPT on this task under zero-shot setting is 83.26%. The data recorded in the table are the best results of ten retests.

## 5 Analysis

In this section, we provide further analysis of the TRF module and DRC module in our proposed method for model extraction attacks.

### 5.1 Impact of Task-Relevant Corpora

We investigate the impact of corpora with varying degrees of relevance to the target task on the performance of model extraction attacks. The experiments are conducted on a simulated victim model trained on IMDB with different query budgets. To assess the influence of the corpus, we replace the default WikiText-103 corpus with other datasets that are more closely related to the victim model's dataset, such as the SST-2 and IMDB training sets. For query sampling, we adopt a uniform random strategy. The results, shown in Table 4 and Table 7, indicate that using the IMDB training data as the corpus yields the best performance, followed

by using the SST-2 training data, both of which outperform the use of WikiText-103. This discrepancy can be attributed to the similarity between the attacker's corpus and the training data distribution of the victim model, i.e., SST-2 and IMDB are both about movie reviews, resulting in a higher data correlation. This observation motivates the Task Relevance Filter in our method, as it aims to filter texts from the public unannotated corpus that closely resemble the data distribution of the task.

### 5.2 Role of Data Reduction based on Clustering

To further explore the role of the clustering-based data reduction technique, we qualitatively visualize the query sampled by DRC and RS on a subset of WikiText-103 with t-SNE (Van der Maaten and Hinton, 2008). Figure 5 depicts the distribution of the sampled data. We can see that the data sampled by DRC displays a broader distribution with larger inter-point distances, whereas the data sampled by RS show some overlapping. This observation demonstrates the capability of DRC to sample data based on their information and effectively reduce redundancy.

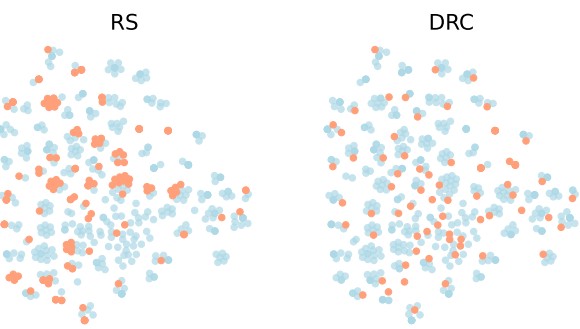

Figure 5: Comparison of data distribution: DRC Sampling vs. RS (Random Sampling). The blue point represents the data pool consisting of the text in the subset of WikiText-103. The selected data points by DRC or RS are marked in red.

| Corpus | × 0.003 | × 0.005 | × 0.008 | × 0.01 | × 0.02 |
|---|---|---|---|---|---|
| WikiText-103 | 53.4 ± 3.8 (63.1) | 59.8 ± 7.8 (71.8) | 63.2 ± 8.0 (73.5) | 63.2 ± 8.3 (75.0) | 66.8 ± 11.1 (78.9) |
| SST-2 training data | 70.8 ± 6.5 (77.4) | 72.3 ± 7.0 (81.0) | 76.5 ± 5.5 (81.7) | 78.9 ± 5.6 (84.2) | 77.4 ± 5.3 (85.3) |
| IMDB training data | 75.5 ± 5.9 (84.2) | 80.0 ± 5.0 (84.8) | 84.7 ± 2.0 (86.6) | 84.8 ± 1.3 (85.6) | 85.6 ± 1.3 (87.2) |

Table 4: Model extraction results (Agreement, %) under different query budgets with a different corpus on IMDB dataset. The results shown in the table are the mean and standard deviation. The max value is shown in green.

| Task | RS example | MeaeQ example |
|---|---|---|
| Hate Speech | After the song was released in the United Kingdom on June 8, 1985, it debuted at number 25 and peaked at number imperial. (**not relevant**) | Lifton describes Mengele as sadistic, lacking empathy, and extremely antisemitic, believing the Jews should be eliminated entirely as an inferior and dangerous race. |
| SST-2 / IMDB | During World War I, she was known as Norman El Brazilian in service with the United States Army and as USS El NHL 50th in service with the United States 1957. (**not relevant**) | Roger Ebert, writing in the Chicago Sun-Times, awarded the film four out of four stars and praised the detail in the portrait of Johnny Marco, saying " Coppola is a fascinating director. |
| AG News | He eventually convinced silence of Powell foreign to the Hollywood and this led to Pokémon being accepted to the Manhattan Project without devices his institutions. | In August 2015, Apple admitted that some iPhone 6 Plus may have faulty cameras that could be causing photos to look blurry and initiated a replacement program. |

Table 5: Query examples sampled by RS and MeaeQ for four tasks, demonstrating that MeaeQ can select samples that are semantically more relevant to the target task. More query examples can be found in Appendix D.

## 5.3 Analysis of High-Frequency Words

We conduct an analysis of the text content in the query set generated by MeaeQ. Specifically, we segment the text in the query set by word and calculate their frequency of occurrence. Figure 6 presents the top 20 most frequent words in the query sets constructed by RS and MeaeQ for the Hate Speech dataset at query budget × 0.5. We observe that the text filtered by MeaeQ contains a higher frequency of words associated with hate speech, such as "hate", "antisemitic", "nazi", "racist", etc. In con-

trast, these words are rarely seen in RS, confirming that MeaeQ effectively filters task-related data and enhances data diversity.

In Table 5, we also provide query examples sampled by RS and MeaeQ, highlighting task-specific words in colors. It is apparent that the content style of the query generated by MeaeQ closely aligns with the target task. For hate speech, they feature negative and hateful sentiment words; for SST-2/IMDB, they contain positive or negative movie reviews; and for AG News, they include key news elements like 'when', 'who', and 'what'. On the contrary, the contents of RS examples are often not relevant to the given task.

## 6 Conclusion

In this paper, we propose a straightforward yet effective model extraction attack method MeaeQ. In particular, we initially utilize a zero-shot sequence inference classifier in combination with the API service information to filter data with high task relevance from a public unannotated text corpus. Subsequently, we employ a clustering-based data reduction technique to obtain representative data as queries for the attack. Extensive experiments demonstrate that our method achieves superior performance than baselines with limited query budgets on four benchmark datasets.

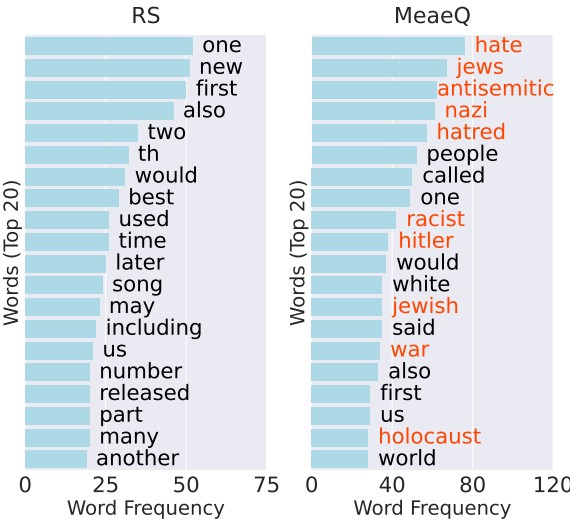

Figure 6: Top-20 frequent words in the query sets constructed by RS and MeaeQ respectively at query budget × 0.5 on Hate Speech.

## Limitations

Our study focuses on model extraction attacks in text classification tasks, in line with existing baselines (Pal et al., 2020; Krishna et al., 2020). The applicability of MeaeQ to other natural language generation tasks, such as machine translation or text summarization, remains unexplored. We are currently investigating model extraction attacks in these more complex tasks.

## Ethics Statement

The data used in this work including WikiText-103 (Merity et al., 2017) and the victim model datasets (de Gibert et al., 2018; Zhang et al., 2015; Socher et al., 2013; Maas et al., 2011) are all open source and do not involve any ethical violations. However, it is essential to note that the method proposed in this paper can be potentially exploited by malicious users to extract real-world black-box models due to its low cost. Therefore, it is crucial to raise awareness within the community regarding the significance of model extraction attacks.

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

# A Results on Accuracy

In this section, we provide the accuracy results of the experimental part in the main text.

## A.1 Main Results

For the main experiment, We list the accuracy results in Table 6. From Table 6, we can see that in all settings, MeaeQ is also better than the baselines in terms of accuracy. Furthermore, we notice that MeaeQ can achieve similar or higher accuracy performance than the baselines while using approximately half of the query budget (as observed in the comparison between MeaeQ of Group A and the baseline method of Group B). Notably, in the best case, for using the query budget set by Group A, the accuracy of the model extracted by MeaeQ can reach 88.8% of the victim model for Hate Speech, 92.8% for SST-2, 86.1% for IMDB, and 91.1% for AG News, which also demonstrate that MeaeQ is an efficient model extraction attack method.

## A.2 Cross Model Extraction

We present the results of cross-architecture model extraction in terms of accuracy in Figure 7. It is evident that MeaeQ consistently outperforms the baseline method in extracting models with higher accuracy across different model architectures. Furthermore, the results on the agreement of MeaeQ, as shown in Figure 2, reveal that the model extracted using BERT exhibits higher functional consistency than RoBERTa, which contradicts the observation in terms of accuracy in Figure 7. This indicates that accuracy and agreement do not necessarily exhibit a complete positive correlation.

## A.3 Ablation Study

The accuracy results of the ablation study are presented in Figure 8. Similar to the observation in the main body, we find that MeaeQ exhibits a significant performance advantage over the two variants at a low query budget.

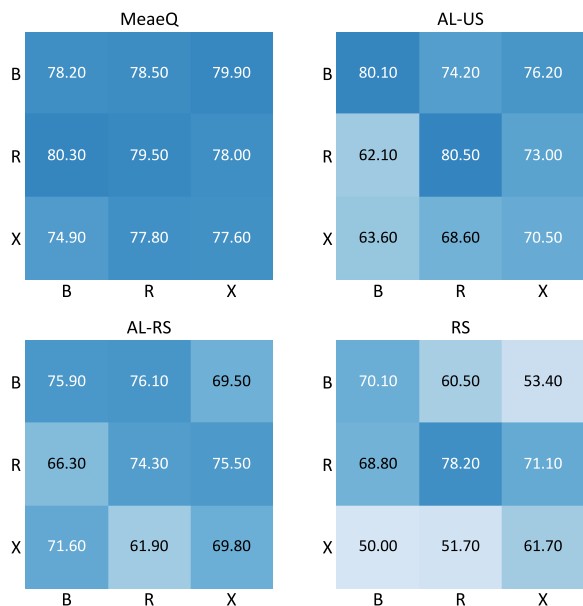

Figure 7: Cross model extraction results (Accuracy, %) on Hate Speech at query budget $\times$ 0.5. The horizontal / vertical axes represent victim / extracted model architecture respectively. 'B', 'R', and 'X' represent $BERT_{Base}$, $RoBERTa_{Base}$, and $XLNet_{Base}$. Darker colors represent larger values of Accuracy.

## A.4 Impact of Task-Relevant Corpora

Table 7 presents the accuracy results obtained using corpora with varying degrees of relevance to the target task. The findings align with the corresponding chapter in the main text, confirming that task-relevant data significantly enhance the performance of model extraction attacks.

# B Details of Datasets

The dataset details are listed in Table 8. We split the original train set into new train and validation sets with a 7:1 ratio for SST-2, 9:1 for IMDB and Hate Speech, and 4:1 for AG News. The original validation set serves as the test set for SST-2, while the others remain unchanged.

# C Hyperparameter Settings

We list the hyperparameters in the main experiment in Tabel 9. Both the victim model and the extracted model use the same training hyperparameters. Other parameters about the model are default parameters from huggingface[6].

---

[6]https://huggingface.co/docs/transformers/index

| Query Budget | Method | Hate Speech ($\times 1 = 1914$) | SST-2 ($\times 1 = 67349$) | IMDB ($\times 1 = 40000$) | AG News ($\times 1 = 120000$) |
|---|---|---|---|---|---|
| - | Victim | 83.89 | 92.55 | 89.04 | 92.43 |
| Group A $\times 0.1$ / $\times 0.003$ (191 / 201 / 120 / 360) | RS | 50.0 ± 0.0 (50.0) | 66.0 ± 8.0 (81.1) | 52.5 ± 3.7 (62.2) | 75.0 ± 3.0 (78.0) |
| | AL-RS | 49.2 ± 2.1 (50.0) | 71.9 ± 6.6 (83.3) | 54.3 ± 4.3 (63.6) | 79.4 ± 2.7 (83.0) |
| | AL-US | 56.6 ± 6.0 (67.2) | 72.2 ± 8.4 (83.7) | 55.4 ± 4.2 (61.9) | 75.0 ± 7.3 (83.1) |
| | MeaeQ | **69.1 ± 5.1** (74.5) | **81.0 ± 3.7** (85.9) | **63.6 ± 8.3** (76.7) | **80.6 ± 2.8** (**84.2**) |
| Group B $\times 0.2$ / $\times 0.005$ (382 / 335 / 200 / 600) | RS | 50.0 ± 0.1 (50.0) | 74.9 ± 6.6 (85.1) | 58.7 ± 7.2 (69.6) | 73.0 ± 9.8 (84.2) |
| | AL-RS | 50.4 ± 2.3 (56.9) | 80.5 ± 4.2 (85.8) | 57.9 ± 7.1 (72.6) | 80.7 ± 3.4 (84.5) |
| | AL-US | 67.1 ± 5.3 (73.2) | 76.8 ± 7.5 (84.2) | 60.8 ± 7.7 (74.2) | 81.2 ± 3.3 (85.4) |
| | MeaeQ | **72.9 ± 2.4** (75.7) | **82.8 ± 2.6** (86.9) | **71.1 ± 4.3** (75.5) | **82.7 ± 1.6** (85.8) |
| Group C $\times 0.3$ / $\times 0.008$ (574 / 536 / 320 / 960) | RS | 51.7 ± 3.6 (61.1) | 77.4 ± 8.3 (86.2) | 61.9 ± 7.4 (71.4) | 80.6 ± 2.8 (84.2) |
| | AL-RS | 54.4 ± 6.9 (67.0) | 82.4 ± 4.2 (87.2) | 58.6 ± 5.7 (67.2) | 78.2 ± 2.0 (80.1) |
| | AL-US | 73.5 ± 4.6 (80.3) | 80.5 ± 5.0 (86.6) | 70.6 ± 5.0 (76.0) | 82.0 ± 0.8 (83.5) |
| | MeaeQ | **74.9 ± 1.2** (77.4) | **84.8 ± 1.7** (87.6) | **76.7 ± 1.6** (78.6) | **83.4 ± 1.5** (85.7) |

Table 6: Model extraction result (Accuracy, %) under different query budgets. $\times 0.1$ / $\times 0.2$ / $\times 0.3$ query budgets are set for Hate Speech dataset. $\times 0.003$ / $\times 0.005$ / $\times 0.008$ query budgets are set for others. The specific number of queries for different datasets is shown in parentheses below the query rates. The results shown in the table are the mean and standard deviation while the max value is shown in green.

| Corpus | $\times 0.003$ | $\times 0.005$ | $\times 0.008$ | $\times 0.01$ | $\times 0.02$ |
|---|---|---|---|---|---|
| WikiText-103 | 52.5 ± 3.7 (62.2) | 58.7 ± 7.2 (69.6) | 61.9 ± 7.4 (71.4) | 61.7 ± 7.6 (72.0) | 65.0 ± 10.2 (75.8) |
| SST-2 training data | 68.5 ± 6.1 (74.9) | 70.9 ± 5.8 (77.8) | 74.2 ± 4.5 (77.9) | 76.2 ± 4.4 (80.3) | 75.0 ± 4.4 (81.3) |
| IMDB training data | 73.3 ± 5.2 (80.7) | 77.0 ± 4.4 (81.1) | 81.1 ± 1.7 (82.5) | 81.2 ± 1.0 (82.5) | 81.6 ± 1.0 (82.6) |

Table 7: Model extraction results (Accuracy, %) under different query budgets with a different corpus on IMDB dataset. The results shown in the table are the mean and standard deviation. The max value is shown in green.

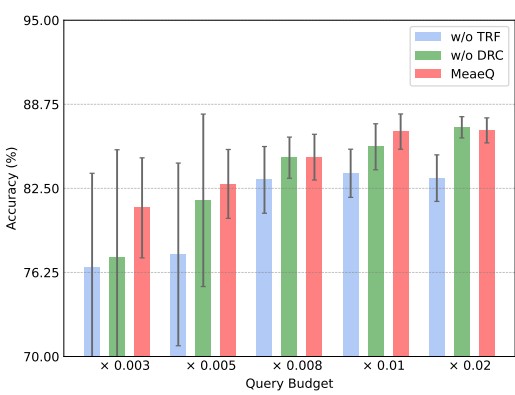

Figure 8: Ablation study results (Accuracy, %) on SST-2 dataset. The horizontal axes represent different query budgets. Error bars represent standard deviation.

## D Query Examples

More query examples are presented in Table 11.

| Datasets | Hate Speech | SST-2 | IMDB | AG News |
|---|---|---|---|---|
| # of class | 2 | 2 | 2 | 4 |
| train size | 1722 | 58930 | 36000 | 96000 |
| validation size | 192 | 8419 | 4000 | 24000 |
| test size | 478 | 872 | 10000 | 7600 |
| total | 2392 | 68221 | 50000 | 127600 |

Table 8: The statistics of the datasets.

| Dataset | Hate Speech | SST-2 | IMDB | AG News |
|---|---|---|---|---|
| adam weight decay | 1e-4 | 1e-4 | 1e-4 | 1e-4 |
| batch size | 32 | 32 | 32 | 16 |
| learning rate | 3e-5 | 3e-5 | 3e-5 | 5e-5 |
| truncate max length | 128 | 128 | 128 | 256 |

Table 9: Details of Hyperparameters in the main experiment only using $BERT_{Base}$ as the model architecture.

| Model Architecture | adam weight decay | batch size | learning rate | truncate max length |
|---|---|---|---|---|
| $XLNet_{Base}$ | 1e-4 | 32 | 2e-5 | 128 |
| $GPT-2_{Small}$ | 1e-5 | 32 | 8e-5 | 128 |
| $GPT-2_{Medium}$ | 1e-5 | 32 | 1e-5 | 128 |
| $RoBERTa_{Base}$ | 1e-4 | 32 | 2e-5 | 128 |
| $RoBERTa_{Large}$ | 1e-4 | 32 | 8e-6 | 128 |

Table 10: Details of Hyperparameters in the experiments using different model architectures on Hate Speech.

| Task | MeaeQ example |
|---|---|
| Hate Speech | Lloyd Paul Stryker, who wrote a highly favorable 1929 biography of Johnson, labeled Stevens as a " horrible old man ... craftily preparing to strangle the bleeding, broken body of the South " and who thought it would be " a beautiful thing " to see " the white men, especially the white women of the South, writhing under negro domination ". |
| | Run by the Abbé Norbert Wallez, the paper described itself as a " Catholic Newspaper for Doctrine and Information " and disseminated a far-right, fascist viewpoint. |
| | One dismissed customer even yells at him, " [ w ] hy don't you go back to your own country ? ", returning the spotlight of racial prejudice on him. |
| | Houston Stewart Chamberlain's work The Foundations of the Nineteenth Century (1900), one of the first to combine Social Darwinism with antisemitism, describes history as a struggle for survival between the Germanic peoples and the Jews, whom he characterized as an inferior and dangerous group. |
| | It talks about permitting arson and destroying Jewish businesses and synagogues, and orders the confiscation of all " archival material " out of Jewish community centres and synagogues. |
| SST-2 / IMDB | Roger Ebert of the Chicago Sun-Times praised the film while comparing it to the original. |
| | Ebert and his colleague, Gene Siskel, gave the film a " Two Thumbs Up " rating on their syndicated television program, Siskel and Ebert and the Movies. |
| | AMC's Chris Cabin criticized the movie, arguing that its director " seems not to have the faintest idea of how to properly approach the subject ", because the film is, in Cabin's view, " unabashedly pro-Palestine ". |
| | BBC film critic Nev Pierce believed the film had spectacular set-pieces, but felt there was no strong narrative arc to keep the viewer interested. |
| | Pittsburgh Post-Gazette critic Ron Weiskind called the film incompetent and criticized the film's acting, lack of suspense, and production values. |
| AG News | By 1904, however, Brazil began to seriously consider upgrading its navy to compete with Argentina and Chile. |
| | In March of that year the News of the World released video footage of Mosley engaged in acts with five consenting women in a scenario that the paper alleged involved Nazi role-playing (an allegation that, though dismissed in court as " no genuine basis ", allegedly " ruined " Mosley's reputation). |
| | On April 4, 2016, Ming was elected into the Naismith Memorial Basketball Hall of Fame. |
| | On 11 November 2011, Iommi, Butler, Osbourne, and Ward announced that they were reuniting to record a new album with a full tour in support beginning in 2012. |
| | In the Hinthada District, torrential rains caused flash flooding that killed 18 people and left 14 others missing. |

Table 11: More examples sampled by MeaeQ from WikiText-103 for four tasks.