# OpenReview forum: "MeaeQ: Mount Model Extraction Attacks with Efficient Queries"
_EMNLP/2023/Conference — EMNLP 2023 Main_

### Official Review · Reviewer_GBYG · 2023-07-22

**Soundness:** 4

**Excitement:**

3: Ambivalent: It has merits (e.g., it reports state-of-the-art results, the idea is nice), but there are key weaknesses (e.g., it describes incremental work), and it can significantly benefit from another round of revision. However, I won't object to accepting it if my co-reviewers champion it.

**Paper Topic And Main Contributions:**

This paper presents a new algorithm for model extraction. Model extraction, like model distillation, is the task of training a student model to mimic a large teacher model (often a black-box API). However unlike model distillation, model extraction does not have access to the teacher model's original training data, and instead has to query the teacher model API for labels. Model extraction is considered a security threat, where the goal of an attacker is to reconstruct a local copy of the teacher model.

The paper presents a new algorithm for model extraction, which is shown to be more query efficient than random sampling and active learning. The proposed algorithm constructs data for model in two steps: 1) data filtering - a large corpus of unlabeled data is first filtered to keep the most relevant data to a downstream task, using the downstream task description and a natural language inference classifier; 2) data clustering - the filtered data is then clustered and representative examples are selected from each cluster as the query dataset; 3) The selected examples are then fed to the black-box teacher model to obtain labels, and the student model is trained on this data.

The paper conducts experiments on a variety of tasks like sentiment classification and hate speech classification, and uses BERT-based APIs as the base model. The paper has several interesting ablation studies and analysis of the proposed approach.

**Reasons To Accept:**

1. This paper studies the problem of model extraction, which is a very relevant problem in the age of large language models which are being released behind black-box APIs (like ChatGPT).

2. The paper presents a simple and intuitive algorithm for query-efficient distillation / model extraction.

3. The proposed approach is evaluated on a variety of tasks, and shows strong performance improvements over competing methods like random sampling and active learning on small query budgets. The authors additionally perform a number of ablation experiments to show the effectiveness of the proposed approach.

**Reasons To Reject:**

1. Model extraction is a very relevant topic in the age of ChatGPT, where several approaches have been introduced to construct instruction tuning datasets automatically [1, 2, 3, 4], in an attempt to "extract" closed-source ChatGPT. This paper would be much more timely and exciting if the methods introduced in this paper are applied to the instrucion tuning setting, and the sample efficiency is compared to techniques like self-instruct [1]. The clustering and filtering approach introduced in this paper could be an interesting way to create instruction diversity, and could potentially lead to better instruction tuned models. However, the paper currently focuses on BERT fine-tuning settings which was the paradigm in 2019, and not used much anymore in state-of-the-art language modeling.

2. In Table 1 (appendix version), the victim models perform much better than the extracted models, and I am guessing this is because of how strict the query budget for the extracted models are. Is such a strict query budget (<1K queries) necessary? API access is quite cheap these days --- for instance, 10K queries to ChatGPT (each with 500 tokens), would cost just 10$ according to the OpenAI API [5]. The paper should justify the strict query budgets by presenting cost estimates from different APIs. This is critical since the main premise of the paper is that it is a "query-efficient" method.

3. I have some concerns about the sequence inference classifier that is used to filter data. It's unclear to me what kind of data passes this filter --- the task of filtering data using the task description seems quite different from the original MNLI distribution. Some examples of the filtered data would be helpful to look at.

4. Minor: A discussion on possible defenses against model extraction would be useful.

[1] - https://arxiv.org/abs/2212.10560
[2] - https://gpt4all.io/reports/GPT4All_Technical_Report_3.pdf
[3] - https://crfm.stanford.edu/2023/03/13/alpaca.html
[4] - https://arxiv.org/abs/2305.15717
[5] - https://openai.com/pricing

**Reproducibility:**

4: Could mostly reproduce the results, but there may be some variation because of sample variance or minor variations in their interpretation of the protocol or method.

**Reviewer Confidence:**

4: Quite sure. I tried to check the important points carefully. It's unlikely, though conceivable, that I missed something that should affect my ratings.

---

> ### Author Rebuttal · Authors · 2023-08-27
>
> ### To Reviewer GBYG:
>
> **Q1:** This paper could be more timely and interesting if the proposed model extraction methods are applied to instruction tuning settings. However, the paper currently focuses on BERT fine-tuning settings which was the paradigm in 2019, and not used much anymore in state-of-the-art language modeling.
>
> **A1:** We highly appreciate your insight. In our future work, we aim to explore the transferability of our approach to constructing instruction data and attempt to perform model extraction on LLMs like ChatGPT (or GPT4) for specific tasks or general tasks. However, our method is tailored for detection tasks such as hate speech detection on social media platforms, where state-of-the-art models are often bidirectional encoders rather than decoder-only models or LLMs [1, 2]. In order to scrutinize the applicability of our method to the prevalent paradigm of language modeling (autoregressive model), we conducted additional model extraction experiments. Refer to the "Additional Experiment" section below for further details.
>
>        Moreover, stringent requirements for model size and inference speed exist in both online and offline detection scenarios. Despite BERT's smaller size compared to LLMs, it performs on par or even better in classification tasks. We believe the traditional BERT fine-tuning paradigm remains relevant in these contexts, and our method still poses a threat to model privacy, enabling the theft of well-performing models at minimal cost.
>
> [1] Kocoń, Jan, et al. "ChatGPT: Jack of all trades, master of none." *Information Fusion* (2023): 101861.
>
> [2] Zhong, Qihuang, et al. "Can ChatGPT Understand Too? A Comparative Study on ChatGPT and Fine-tuned BERT." *arXiv preprint arXiv:2302.10198* (2023).
>
> **Additional Experiment**
>
> We conduct model extraction attacks on the ChatGPT-based victim model simulated on the hate speech detection task using manual instructions. The instruction template is shown in Table 1. We set up several types of extracted models with different parameters and architectures, including GPT-2 and RoBERTa. The results of the zero-shot ChatGPT model extraction on the Hate Speech task are reported in Tables 2 and 3. We can find that our method still performs well even though the victim model is ChatGPT-based and the extracted model is autoregressive-based language model, e.g. GPT-2. For instance, GPT-2-small achieves **79.5%** functional similarity (Agreement) and 75.3% test accuracy (**90.4%** of the victim model). We also observe that the RoBERTa series of models outperforms the GPT-2 series in this detection task. Just as the masked language model can be used to extract classification models, it is sensible to use autoregressive models to extract generative models.
>
> **Table 1: Instruction Template**
>
> | Task | Instruction |
> | --- | --- |
> | Hate Speech | You are now working as a hate speech detector. Next, I will give you [batch\_size] sentences. You need to indicate whether it contains hate speech. If it contains hate speech, output "Hate", otherwise output "Nohate". The required format is one output per line and the beginning of each output is numbered sequentially. Sentences:\n 1. [sentence\_1]\n 2. [sentence\_2]\n... |
>
> ###### Detail: The [batch_size] in the instruction represents the number of queries, and [sentence_1], [sentence_2], etc. represent the specific query content.
>
> **Table 2: Model Extraction Results (Agreement, %)**
>
> | Method | GPT-2 (small,117M) | GPT-2 (medium,345M) | RoBERTa (Base, 125M) | RoBERTa (Large, 355M) |
> | --- | --- | --- | --- | --- |
> | RS  | 43.7 | 43.9 | 51.9 | 59.8 |
> | AL-RS | 43.9 | 48.9 | 43.7 | 54.2 |
> | AL-US | 43.9 | 45.6 | 57.3 | 76.6 |
> | MeaeQ (ours) | **79.5** | **75.3** | **82.4** | **85.4** |
>
> ###### Detail: Table 2 shows the model extraction results (Agreement, %) on ChatGPT simulated on the Hate Speech dataset at query budget × 0.8 (about 1531 queries). The data recorded in the table are the best results of ten retests.
>
> **Table 3: Model Extraction Results (Accuracy, %)**
>
> | Method | GPT-2 (small,117M) | GPT-2 (medium,345M) | RoBERTa (Base, 125M) | RoBERTa (Large, 355M) |
> | --- | --- | --- | --- | --- |
> | RS  | 50.0 | 50.2 | 55.6 | 64.4 |
> | AL-RS | 50.2 | 53.6 | 50.0 | 56.3 |
> | AL-US | 50.2 | 50.6 | 60.7 | 73.6 |
> | MeaeQ (ours) | **75.3** | **71.5** | **77.4** | **79.5** |
>
> ###### Detail: Table 3 shows the model extraction results (Accuracy, %) on ChatGPT simulated on the Hate Speech dataset at query budget × 0.8 (about 1531 queries). The accuracy of ChatGPT on this task under the zero-shot setting is 83.26%. The data recorded in the table are the best results of ten retests.
>
> **Q2:** In Table 1 (appendix version), the victim models perform much better than the extracted models, and I am guessing this is because of how strict the query budget for the extracted models are. Is such a strict query budget (<1K queries) necessary? API access is quite cheap these days --- for instance, 10K queries to ChatGPT (each with 500 tokens), would cost just 10$ according to the OpenAI API [5]. The paper should justify the strict query budgets by presenting cost estimates from different APIs. This is critical since the main premise of the paper is that it is a "query-efficient" method.
>
> **A2:** Your insightful question has prompted us to reflect deeply on this matter. Allow us to elaborate on why strict query budgets are indeed necessary.
>
> 1. The performance gain brought to the extracted model by more queries is limited. From the first row of Figure 3 (main text), we can find that beyond 1K queries (e.g., query budget > 0.008 on AG News), the performance of the extracted model plateaus and maintains a certain performance gap with the victim model. We believe this performance gap does not solely stem from the strict query budget, but rather originates from the inherent differences in training data distributions between the attacker and victim models.
>
> 2. Excessive API calls could trigger defense mechanisms, diminishing the success rate of model extraction. Notably, prior work [1, 2, 3] has shown the effectiveness of distinguishing legitimate users from malicious actors through the detection of poorly crafted or excessive queries. Given that, our paper strives to craft high-quality queries within limited budgets to extract a high-performance model successfully.
>
>
> Thanks again for your insightful inquiry. We will delve deeper into justifying the strict query budgets in the revised paper.
>
> [1] Zhang, Zhanyuan, Yizheng Chen, and David Wagner. "SEAT: Similarity Encoder by Adversarial Training for Detecting Model Extraction Attack Queries." *Proceedings of the 14th ACM Workshop on Artificial Intelligence and Security*. 2021.
>
> [2] Chen, Steven, Nicholas Carlini, and David Wagner. "Stateful Detection of Black-Box Adversarial Attacks." _Proceedings of the 1st ACM Workshop on Security and Privacy on Artificial Intelligence_. 2020.
>
> [3] Kariyappa, Sanjay, and Moinuddin K. Qureshi. "Defending Against Model Stealing Attacks with Adaptive Misinformation." _Proceedings of the IEEE/CVF Conference on Computer Vision and Pattern Recognition_. 2020.
>
> **Q3:** I have some concerns about the sequence inference classifier that is used to filter data. It's unclear to me what kind of data passes this filter --- the task of filtering data using the task description seems quite different from the original MNLI distribution. Some examples of the filtered data would be helpful to look at.
>
> **A3:** Thank you for your constructive suggestion. In Section 5.3, we provide examples of high-frequency words that demonstrate the filter's alignment with the task description. For instance, in the hate speech detection task, the filter selects data containing words like "hate", "racist", "war", etc. This analysis illustrates that the filtered data aligns with the task description, validating the effectiveness of the sequence inference classifier, even when the MNLI dataset distribution differs from the corpus. We also provide illustrative examples of filtered data in Table 3 below this response, to offer a clearer understanding of the filter’s functionality, which will be added to our revised paper. Thanks again sincerely for your useful suggestion.
>
> **Table 3: Some data examples filtered by the sequence inference classifier**
>
> | Examples for Hate Speech |
> | --- |
> | 1. Lifton describes Mengele as sadistic, lacking empathy, and extremely antisemitic, believing the Jews should be eliminated entirely as an inferior and dangerous race. |
> | 2. Lloyd Paul Stryker, who wrote a highly favorable 1929 biography of Johnson, labeled Stevens as a " horrible old man ... craftily preparing to strangle the bleeding, broken body of the South " and who thought it would be " a beautiful thing " to see " the white men, especially the white women of the South, writhing under negro domination ". |
> | 3. Run by the Abbé Norbert Wallez, the paper described itself as a " Catholic Newspaper for Doctrine and Information " and disseminated a far-right, fascist viewpoint. |
>
> | Examples for SST-2 / IMDB |
> | --- |
> | 1. Roger Ebert of the Chicago Sun-Times praised the film while comparing it to the original. |
> | 2. Ebert and his colleague, Gene Siskel, gave the film a " Two Thumbs Up " rating on their syndicated television program, Siskel and Ebert and the Movies. |
> | 3. Roger Ebert, writing in the Chicago Sun-Times, awarded the film four out of four stars and praised the detail in the portrait of Johnny Marco, saying " Coppola is a fascinating director. |
>
> | Examples for AG News |
> | --- |
> | 1. By 1904, however, Brazil began to seriously consider upgrading its navy to compete with Argentina and Chile. |
> | 2. In March of that year the News of the World released video footage of Mosley engaged in acts with five consenting women in a scenario that the paper alleged involved Nazi role-playing (an allegation that, though dismissed in court as " no genuine basis ", allegedly " ruined " Mosley's reputation). |
> | 3. On April 4, 2016, Ming was elected into the Naismith Memorial Basketball Hall of Fame. |
>
> **Q4:** A discussion on possible defenses against model extraction would be useful.
>
> **A4:** We appreciate your insightful suggestion. Exploring possible defenses against model extraction is indeed a valuable avenue for discussion. Here are several potential defense strategies that we have considered:
>
> 1. One potential defense strategy could involve training a membership classifier to discern whether a user's input originates from genuine data distribution or from out-of-distribution. While the Task Relevance Filter identifies task-related data, it might capture only partial word associations. The membership classifier would need to grasp more specific patterns related to the task, such as text formatting, logical coherence, or contextual consistency. As attackers use individual sentences as queries, the isolated presentation of a sentence could be perplexing due to the possible absence of context.
>
> 2. Another conceivable defense approach entails introducing perturbations to the API response, possibly altering labels from positive to negative examples; however, this might adversely affect legitimate user inputs.
>
> 3. A third defense strategy could combine the previous two by leveraging the confidence scores (or probabilities) provided by the membership classifier to determine the degree of perturbation to introduce into API responses.
>
>
> In the revised paper, we will add a disscussion on possible defense strategies against model extraction attack.

---

### Official Review · Reviewer_ymJe · 2023-08-01

**Typos Grammar Style And Presentation Improvements:** It would be better if some punctuatio…
**Soundness:** 3

**Excitement:**

3: Ambivalent: It has merits (e.g., it reports state-of-the-art results, the idea is nice), but there are key weaknesses (e.g., it describes incremental work), and it can significantly benefit from another round of revision. However, I won't object to accepting it if my co-reviewers champion it.

**Missing References:**

None

**Paper Topic And Main Contributions:**

The paper introduces MeaeQ (Model extraction attack with efficient Queries), a straightforward and effective method to address certain issues. The method involves using a zero-shot sequence inference classifier, along with API service information, to filter task-relevant data from a public text corpus instead of using a problem domain-specific dataset. Additionally, they utilize a clustering-based data reduction technique to obtain representative data as queries for the attack. Extensive experiments conducted on four benchmark datasets demonstrate that MeaeQ achieves higher functional similarity to the victim model compared to baselines while requiring fewer queries.

**Questions For The Authors:**

Why not try LLMs like GPT and LLAMA instead of BERT?

**Reasons To Accept:**

1. The risk of model extraction attack is important, so the research reported in this paper is practical.

2. This paper is easy to read and understand.

3. The experimental results show the effectiveness of the proposed method.

**Reasons To Reject:**

1. The technical quality is relatively limited in my opinion, just some clustering and filtering methods.

2. The models to be extracted are PLMs like BERT, why not try LLMs like GPT and LLAMA?

3. The proposed method relies on predefined tasks, which cannot be applied to AGI models.

4. Only BERT model is used in experiments.

**Reproducibility:**

3: Could reproduce the results with some difficulty. The settings of parameters are underspecified or subjectively determined; the training/evaluation data are not widely available.

**Reviewer Confidence:**

3: Pretty sure, but there's a chance I missed something. Although I have a good feel for this area in general, I did not carefully check the paper's details, e.g., the math, experimental design, or novelty.

---

> ### Author Rebuttal · Authors · 2023-08-27
>
> ### To Reviewer ymJe:
>
> **Q0:** It would be better if some punctuation can be added to equations.
>
> **A0:** Thank you for your feedback regarding punctuation in equations. We appreciate your suggestion and will make sure to review and enhance the punctuation in equations to improve their clarity and presentation.
>
> **Q1:** The technical quality is relatively limited in my opinion, just some clustering and filtering methods.
>
> **A1:** We acknowledge the limitations of the technical aspects presented in our method. However, it's important to note that the crux of our contribution lies in optimizing query sampling from two angles: query-task relevance and inter-query information redundancy. Our experimental results show that the attacker can extract a replica model with more than 80% functional consistency with the victim model with only a few queries. Despite its apparent simplicity, our method's empirical performance speaks to its effectiveness, aligning with the empirical spirit of EMNLP.
>
> **Q2:** Why not try LLMs like GPT and LLAMA instead of BERT?
>
> **A2:** We value your input on this matter. Although our experiments focused on BERT as a representative example, we acknowledge that LLMs like GPT and LLAMA are significant in the context of model extraction. Therefore, we conducted additional model extraction experiments using GPT series of models as the primary setting to validate the effectiveness of our method. We will include the results of this experiment in the revised version of the paper. Currently, we present the experiment as follows.
>
> **Additional Experiment**
>
> We conduct model extraction attacks on the ChatGPT-based victim model simulated on the hate speech detection task using manual instructions. The instruction template is shown in Table 1. We set up several types of extracted models with different parameters and architectures, including GPT-2 and RoBERTa. The results of the zero-shot ChatGPT model extraction on the Hate Speech task are reported in Tables 2 and 3. We can find that even though the victim model is ChatGPT-based, our method still performs well for both GPT-like and BERT-like models. For instance, RoBERTa-large achieves **85.4%** functional similarity (Agreement) and 79.5% test accuracy (**95.5%** of the victim model); GPT-2-small achieves **79.5%** functional similarity (Agreement) and 75.3% test accuracy (**90.4%** of the victim model). We also observe that the RoBERTa series of models outperforms the GPT-2 series in this detection task. Just as masked language models can be used to extract classification models, it is sensible to use autoregressive models to extract generative models.
>
> **Table 1: Instruction Template**
>
> | Task | Instruction |
> | --- | --- |
> | Hate Speech | You are now working as a hate speech detector. Next, I will give you [batch\_size] sentences. You need to indicate whether it contains hate speech. If it contains hate speech, output "Hate", otherwise output "Nohate". The required format is one output per line and the beginning of each output is numbered sequentially. Sentences:\n 1. [sentence\_1]\n 2. [sentence\_2]\n... |
>
> ###### Detail: The [batch_size] in the instruction represents the number of queries, and [sentence_1], [sentence_2], etc. represent the specific query content.
>
> **Table 2: Model Extraction Results (Agreement, %)**
>
> | Method | GPT-2 (small,117M) | GPT-2 (medium,345M) | RoBERTa (Base, 125M) | RoBERTa (Large, 355M) |
> | --- | --- | --- | --- | --- |
> | RS  | 43.7 | 43.9 | 51.9 | 59.8 |
> | AL-RS | 43.9 | 48.9 | 43.7 | 54.2 |
> | AL-US | 43.9 | 45.6 | 57.3 | 76.6 |
> | MeaeQ (ours) | **79.5** | **75.3** | **82.4** | **85.4** |
>
> ###### Detail: Table 2 shows the model extraction results (Agreement, %) on ChatGPT simulated on the Hate Speech dataset at query budget × 0.8 (about 1531 queries). The data recorded in the table are the best results of ten retests.
>
> **Table 3: Model Extraction Results (Accuracy, %)**
>
> | Method | GPT-2 (small,117M) | GPT-2 (medium,345M) | RoBERTa (Base, 125M) | RoBERTa (Large, 355M) |
> | --- | --- | --- | --- | --- |
> | RS  | 50.0 | 50.2 | 55.6 | 64.4 |
> | AL-RS | 50.2 | 53.6 | 50.0 | 56.3 |
> | AL-US | 50.2 | 50.6 | 60.7 | 73.6 |
> | MeaeQ (ours) | **75.3** | **71.5** | **77.4** | **79.5** |
>
> ###### Detail: Table 3 shows the model extraction results (Accuracy, %) on ChatGPT simulated on the Hate Speech dataset at query budget × 0.8 (about 1531 queries). The accuracy of ChatGPT on this task under the zero-shot setting is 83.26%. The data recorded in the table are the best results of ten retests.
>
> **Q3:** The proposed method relies on predefined tasks, which cannot be applied to AGI models.
>
> **A3:** We acknowledge your concern regarding the applicability of our method to AGI models. Our paper primarily focuses on extracting black-box models for classification tasks rather than generative tasks. While AGI models like ChatGPT demonstrate impressive performance across a range of open-ended tasks, masked language models like BERT still remain state-of-the-art representatives for specific classification tasks [1, 2]. Thus, our method retains much value in extracting black-box models powering classification task's APIs.
>
> Furthermore, we do not believe that our method is inherently incompatible with AGI models or LLMs. In the era of such models, numerous efforts have emerged, utilizing the extraction data from ChatGPT to train the LLaMa by instruction tuning (e.g., Alpaca , Vicuna). As Reviewer GBYG commented, "The clustering and filtering approach introduced in this paper could be an interesting way to create instruction diversity, and could potentially lead to better instruction tuned models." Our method holds potential value in filtering high-quality instruction data, which is also part of our future work.
>
> For instance, we can leverage the self-instruct [3] technique to acquire an extensive and diverse set of instructions. Subsequently, we can apply filters to obtain task-relevant subsets of instructions, e.g., using finance-related instructions for tuning financial LLM[4]. Additionally, the application of clustering methods can be employed to effectively reduce redundant instruction information, thereby further enhancing the quality of instructions. It's important to note that more instructions are not necessarily better [5]; the quality of instructions is of paramount importance [6].
>
> [1] Kocoń, Jan, et al. "ChatGPT: Jack of all trades, master of none." *Information Fusion* (2023): 101861.
>
> [2] Zhong, Qihuang, et al. "Can ChatGPT Understand Too? A Comparative Study on ChatGPT and Fine-tuned BERT." *arXiv preprint arXiv:2302.10198* (2023).
>
> [3] Wang, Yizhong, et al. "SELF-INSTRUCT: Aligning Language Model with Self-Generated Instructions." _arXiv preprint arXiv:2212.10560_ (2022).
>
> [4] Wu, Shijie, et al. "BloombergGPT: A Large Language Model for Finance." _arXiv preprint arXiv:2303.17564_ (2023).
>
> [5] Zhou, Chunting, et al. "LIMA: Less is More for Alignment." _arXiv preprint arXiv:2305.11206_ (2023).
>
> [6] Wei, Lai, et al. "InstructionGPT-4: A 200-Instruction Paradigm for Fine-Tuning MiniGPT-4." _arXiv preprint arXiv:2308.12067_ (2023).
>
> **Q4:** Only BERT model is used in experiments.
>
> **A4:** In fact, we thoroughly explore our method's applicability across model architectures in Section 4.4, where we consider BERT, RoBERTa, and XLNet for both victim and attacker models. The experiment results consistently support its effectiveness across diverse model architectures. In the revised version, we will pay closer attention to detailed descriptions to ensure a more comprehensive presentation of our experimental setup.

---

### Official Review · Reviewer_rQZ7 · 2023-08-21

**Soundness:** 4

**Excitement:**

4: Strong: This paper deepens the understanding of some phenomenon or lowers the barriers to an existing research direction.

**Paper Topic And Main Contributions:**

The paper provides an effective extraction method based on two filtering processes: selecting the highly relevant samples and reducing the number of samples to be queried.

**Reasons To Accept:**

- A targetted method for extraction, not similar to the previous method, is based on random sampling, which increases the success rate and lowers the required number of queries

- Outperforms the previous baselines.

- Well-written, and the experiments and ablations are informative.

**Reasons To Reject:**

- Employing the method on various nlp tasks or classification tasks(e.g. offensive detection) would be better to confirm the efficiency of the method.
- Even though MLM is important and used in many applications, the method should also be employed in the autoregressive language models, which are used more in real-world applications

**Reproducibility:**

3: Could reproduce the results with some difficulty. The settings of parameters are underspecified or subjectively determined; the training/evaluation data are not widely available.

**Reviewer Confidence:**

3: Pretty sure, but there's a chance I missed something. Although I have a good feel for this area in general, I did not carefully check the paper's details, e.g., the math, experimental design, or novelty.

---

> ### Author Rebuttal · Authors · 2023-08-27
>
> ### To Reviewer rQZ7:
>
> **Q1:** Employing the method on various NLP tasks or classification tasks (e.g. offensive detection) would be better to confirm the efficiency of the method.
>
> **A1:** Thank you for your suggestion. We fully recognize the importance of evaluating our method across a broader spectrum of NLP tasks. Therefore, in our experiments, we have already simulated a range of detection tasks, such as SST-2 for short text sentiment analysis, IMDB for sentiment classification in long texts, AG News for news text classification, and Hate Speech for hate content detection on social media platforms. The results highlight the effectiveness of our approach over baselines, reinforcing its notable impact. Also, in the final version of the paper, we will add more classification tasks (e.g. OffensEval 2019 [1] for offensive language identification and Enron Spam [2] for spam email detection ) to confirm the efficiency of our method.
>
> [1] Zampieri, Marcos, et al. "Semeval-2019 Task 6: Identifying and Categorizing Offensive Language in Social Media (OffensEval)." *arXiv preprint arXiv:1903.08983* (2019).
>
> [2] Metsis, Vangelis, Ion Androutsopoulos, and Georgios Paliouras. "Spam Filtering with Naive Bayes-Which Naive Bayes?." *CEAS*. Vol. 17. 2006.
>
>
>
> **Q2:** Even though MLM is important and used in many applications, the method should also be employed in the autoregressive language models, which are used more in real-world applications.
>
> **A2:** We fully agree with your point. In fact, we have systematically conducted model extraction experiments on autoregressive language models. Due to the page limit of the paper, we did not include this experiment. In the final version of the paper, we will supplement this experiment. Now we report the experiment as follows.
>
> ####
>
> **Additional Experiment**
>
> We conduct model extraction attacks on the ChatGPT-based victim model simulated on the hate speech detection task using manual instructions. The instruction template is shown in Table 1. We set up several types of extracted models with different parameters and architectures, including GPT-2 and RoBERTa. The results of the zero-shot ChatGPT model extraction on the hate speech detection task are reported in Tables 2 and 3. We can find that our method still performs well even though the victim model is ChatGPT-based and the extracted model is autoregressive-based language model, e.g. GPT-2. For instance, GPT-2-small achieves **79.5%** functional similarity (Agreement) and 75.3% test accuracy (**90.4%** of the victim model). We also observe that the RoBERTa series of models outperforms the GPT-2 series in this detection task. Just as MLM can be used to extract classification models, it is sensible to use autoregressive models to extract generative models.
>
>
>
> **Table 1: Instruction Template**
>
> | Task        | Instruction                                                  |
> | ----------- | ------------------------------------------------------------ |
> | Hate Speech | You are now working as a hate speech detector. Next, I will give you [batch\_size] sentences. You need to indicate whether it contains hate speech. If it contains hate speech, output "Hate", otherwise output "Nohate". The required format is one output per line and the beginning of each output is numbered sequentially. Sentences:\n1. [sentence\_1]\n2. [sentence\_2]\n... |
>
> ###### Detail: The [batch_size] in the instruction represents the number of queries, and [sentence_1], [sentence_2], etc. represent the specific query content.
>
>
>
> **Table 2: Model Extraction Results (Agreement, %)**
>
> | Method       | GPT-2 (small,117M) | GPT-2 (medium,345M) | RoBERTa (Base, 125M) | RoBERTa (Large, 355M) |
> | ------------ | ------------------ | ------------------- | -------------------- | --------------------- |
> | RS           | 43.7               | 43.9                | 51.9                 | 59.8                  |
> | AL-RS        | 43.9               | 48.9                | 43.7                 | 54.2                  |
> | AL-US        | 43.9               | 45.6                | 57.3                 | 76.6                  |
> | MeaeQ (ours) | **79.5**           | **75.3**            | **82.4**             | **85.4**              |
>
> ###### Detail: Table 2 shows the model extraction results (Agreement, %) on ChatGPT simulated on the Hate Speech dataset at query budget × 0.8 (about 1531 queries). The data recorded in the table are the best results of ten retests.
>
>
>
> **Table 3: Model Extraction Results (Accuracy, %)**
>
> | Method       | GPT-2 (small,117M) | GPT-2 (medium,345M) | RoBERTa (Base, 125M) | RoBERTa (Large, 355M) |
> | ------------ | ------------------ | ------------------- | -------------------- | --------------------- |
> | RS           | 50.0               | 50.2                | 55.6                 | 64.4                  |
> | AL-RS        | 50.2               | 53.6                | 50.0                 | 56.3                  |
> | AL-US        | 50.2               | 50.6                | 60.7                 | 73.6                  |
> | MeaeQ (ours) | **75.3**           | **71.5**            | **77.4**             | **79.5**              |
>
> ###### Detail: Table 3 shows the model extraction results (Accuracy, %) on ChatGPT simulated on the Hate Speech dataset at query budget × 0.8 (about 1531 queries). The accuracy of ChatGPT on this task under the zero-shot setting is 83.26%. The data recorded in the table are the best results of ten retests.

---

### Meta-Review · Area_Chair_TMd3 · 2023-09-24

**Recommendation:** 4

**Metareview:**

The paper presents an innovative and practically relevant approach to model extraction supported by strong experimental evidence. However, it could improve by extending experiments to include larger language models, and a more comprehensive range of tasks would further enhance its quality and applicability. Additionally, considering potential technical depth improvements could strengthen its overall contribution.

---

### Decision · Program_Chairs · 2023-10-07

**Decision:**

Accept-Main

**Comment:**

The paper presents an innovative and practically relevant approach to model extraction supported by strong experimental evidence. However, it could improve by extending experiments to include larger language models, and a more comprehensive range of tasks would further enhance its quality and applicability. Additionally, considering potential technical depth improvements could strengthen its overall contribution.